# Commensal Intestinal Protozoa—Underestimated Members of the Gut Microbial Community

**DOI:** 10.3390/biology11121742

**Published:** 2022-11-30

**Authors:** Magdalena Dubik, Bartosz Pilecki, Jesper Bonnet Moeller

**Affiliations:** 1Department of Cancer and Inflammation Research, Institute of Molecular Medicine, University of Southern Denmark, 5000 Odense, Denmark; 2Danish Institute for Advanced Study, University of Southern Denmark, 5230 Odense, Denmark

**Keywords:** bacteria, *Blastocystis*, *Dientamoeba*, *Entamoeba*, gut immunity, inflammatory bowel disease, intestinal protozoa, microbiota, SCFA

## Abstract

**Simple Summary:**

Gut-associated protozoa are a heterogenous group of microbes that frequently reside within humans. The role of commensal intestinal protozoa and their interaction with the human organism is a highly complex topic, and an area of research that has been largely neglected. We argue that some protozoa species might be beneficial inhabitants of the gut that can significantly impact human health and disease. Here, we aimed to comprehensively review existing literature on the conflicting outcomes of protozoa colonization, their interaction with the bacterial microbiota, and the crosstalk between the protozoa and the host immune system. Moreover, we emphasize the importance for future studies to investigate these aspects of protozoa colonization that will undoubtedly increase our understanding of complex interactions between intestinal protozoa, other microbiota organisms, and the human host.

**Abstract:**

The human gastrointestinal microbiota contains a diverse consortium of microbes, including bacteria, protozoa, viruses, and fungi. Through millennia of co-evolution, the host–microbiota interactions have shaped the immune system to both tolerate and maintain the symbiotic relationship with commensal microbiota, while exerting protective responses against invading pathogens. Microbiome research is dominated by studies describing the impact of prokaryotic bacteria on gut immunity with a limited understanding of their relationship with other integral microbiota constituents. However, converging evidence shows that eukaryotic organisms, such as commensal protozoa, can play an important role in modulating intestinal immune responses as well as influencing the overall health of the host. The presence of several protozoa species has recently been shown to be a common occurrence in healthy populations worldwide, suggesting that many of these are commensals rather than invading pathogens. This review aims to discuss the most recent, conflicting findings regarding the role of intestinal protozoa in gut homeostasis, interactions between intestinal protozoa and the bacterial microbiota, as well as potential immunological consequences of protozoa colonization.

## 1. Introduction

The mammalian gut harbors a vast number of viruses, bacteria, fungi, and protozoa (single-celled eukaryotes), collectively referred to as the microbiota [1]. Dynamic interplay between these distinct microbiota constituents and the host contributes to many essential physiological processes, including development, metabolism, and immunity [2,3]. It is becoming increasingly evident that disruption of this complex ecosystem, referred to as gut dysbiosis, contributes to the development of various gastrointestinal as well as systemic diseases, such as inflammatory bowel disease (IBD), metabolic disorder, autoimmunity, and cancer [4,5,6]. Furthermore, emerging evidence of a bidirectional crosstalk between the intestinal microbiota and the brain has linked dysbiosis to various diseases of the central nervous system, such as depression, multiple sclerosis, and Parkinson’s disease [7,8]. It is well recognized that the intestinal microbiota plays a pivotal role in the manifestations of IBD [9]. IBD is a chronic inflammatory disease of the intestinal lining that can be classified into two distinct conditions, Crohn’s disease or ulcerative colitis [10]. Although usually not fatal, IBD is associated with lowered life expectancy and significantly decreased quality of life for patients that suffer from chronic symptoms, including persistent diarrhea, abdominal pain, and rectal bleeding. Moreover, chronic inflammation in IBD has been associated with serious, often fatal comorbidities, including cancer, cardiovascular diseases, and liver diseases [11,12]. The rate of incidence of IBD has been rising dramatically since the industrial revolution, with a current global burden of more than 6 million people [13]. Although the IBD etiology remains largely unknown, it has been hypothesized that the industrial lifestyle, which includes increased use of antibiotics and a diet rich in highly processed foods, has resulted in detrimental changes to the intestinal microbiota that significantly contribute to disease risk [9,14]. Therefore, a detailed understanding of the biological roles of distinct intestinal microbial groups, their mutual interactions, as well as their impact on human disease is of paramount importance.

Since most studies have concentrated on gut-resident bacteria as the main component of the microbiota, the mechanisms and consequences of intestinal protozoa colonization have only recently begun to be clarified [3,15,16,17]. Protozoa are a diverse group of single-celled eukaryotic organisms that can be found in a variety of environments either as free-living or parasitic/symbiotic microbes. Historically, protozoa have been classified into four subgroups: amoebas, flagellates, coccidians, and ciliates, and their categorization depends on specific morphological features, such as internal structure and motility [18]. After the emergence of molecular phylogenetics, an updated classification has been proposed, integrating insights from the genomic studies with the structural and biochemical evidence. Thus, protozoa have been proposed to comprise two subkingdoms, with Choanozoa and Amoebozoa grouped together as the subkingdom Sarcomastigota, while Alveolata, Rhizaria, Excavata, and Apusozoa constitute the subkingdom Biciliata [19]. More recently, an even more updated classification has been proposed [20]. From an evolutionary point of view, eukaryotic microbes such as protozoa have co-evolved with humans and undoubtedly affected the dynamics of the gut microbiota [21]. Despite extensive strides in parasitic research, including studies of pathogenic protozoa, the role of commensal protozoa in shaping the immune landscape of the gut remains enigmatic and questioned [22,23,24]. One of the main challenges lies in the characteristic features and biological classification of commensal protozoa [25]. By definition, commensal microbes reside within the host without causing a negative health impact and are well tolerated by the immune system [26]. However, due to the highly dynamic nature of the host–microbiota interactions, a particular protozoa can be classified as commensal rather than parasitic, and vice-versa, often in a context-specific manner [21,27,28]. Furthermore, heterogeneity in experimental design, differences between protozoa species, and geographical changes in gut microbiota all result in a lack of consensus regarding the exact role of intestinal protozoa and their contribution to mucosal immune homeostasis [15,29]. It is well established that the bacterial compartment of human gut microbiota comprises a plethora of different species, ranging from beneficial to opportunistic and/or pathogenic [30]. We hypothesize that a similar paradigm also exists for intestinal protozoa that frequently inhabit the human gastrointestinal tract. From this perspective, we aim here to comprehensively review existing evidence of commensal and potentially beneficial aspects of protozoa colonization in human health and disease.

## 2. Intestinal Protozoa—Pathogenic, Commensal, or Beneficial?

*Giardia lamblia*, *Entamoeba histolytica,* and *Cryptosporidium* spp. are among the most common enteric protozoa parasites that significantly contribute to acute gastroenteritis and diarrheal disease globally [31,32]. Their virulence factors and intestinal invasion mechanisms are well characterized and have been frequently reviewed [33,34,35,36]. Besides gastrointestinal manifestations, infections with these protozoa can lead to serious health-threatening conditions, such as amoebic liver abscess and amoebic colitis [36]. However, while parasitic infections still pose a great health burden, numerous intestinal protozoa species are considered non-pathogenic (Table 1) [29,37]. Lokmer et al. applied a metagenomic approach to examine the occurrence of commensal and potentially beneficial protozoa and to study their ecological dynamics in several worldwide populations. They showed a high prevalence of *Blastocystis* spp., *Entamoeba* spp., and various other protozoa genera in healthy individuals across human populations [37]. Other epidemiological studies have shown that species such as *Dientamoeba* and *Enteromonas* are common residents of the human gut, at least in some parts of the world [37,38].

### 2.1. Blastocystis spp.

*Blastocystis* spp. is one of the most prevalent protozoa found in humans, with an estimated 1 billion colonized individuals worldwide [22,40]. Historically, *Blastocystis* was predominantly characterized as a parasitic protozoa [53], but conflicting results regarding its pathogenic potential and clinical significance have emerged in several studies [40]. *Blastocystis* has been associated with the etiology of irritable bowel syndrome (IBS) [54,55,56,57] and IBD [58]. In contrast, other cohort studies have found no correlation between gastrointestinal symptoms and the presence of *Blastocystis,* either in healthy subjects or IBS patients [59,60]. Similarly, the prevalence of *Blastocystis* infection has been inconsistently reported to be higher in either immunocompetent or immunocompromised individuals depending on the study [61,62]. One possible explanation for this discrepancy is that *Blastocystis* has mainly been investigated as a causative agent in disease propagation, with limited information about its distribution in a healthy population. With recent advances in sequencing technologies and an increased number of epidemiological surveys, it has become evident that the presence of Blastocystis is a common occurrence in both healthy and symptomatic individuals, which inevitably questions its alleged pathogenicity [63,64]. To date, at least 17 *Blastocystis* subtypes (ST) have been identified, of which nine are found in humans (ST1-ST9), with ST1–ST4 accounting for up to 90% of all occurrences [40,65]. Recent findings suggest an association between *Blastocystis* subtype variation and pathogenic potential. For instance, ST2 was suggested to be a non-pathogenic type [66], while ST3 has frequently been identified as a major subtype among patients with gastrointestinal disorders [67,68]. A study by Ali et al. showed a significant association between *Blastocystis* subtypes and colorectal cancer (CRC) [69]. Although a similar prevalence of *Blastocystis* ST1, 2, and 3 were observed between CRC and non-cancer individuals, a rare ST7 was identified in CRC patients [69].

At the other end of the spectrum, beneficial roles for *Blastocystis* have also been proposed. Colonization with *Blastocystis* is associated with higher microbial diversity and richness, both of which are suggested to benefit intestinal health [22,70]. Additionally, it has been shown that body mass index is strongly negatively correlated with *Blastocystis* presence [71]. Several studies have reported that colonization with *Blastocystis* is more common in healthy subjects than in patients with active IBD, IBS, or CRC, supporting that *Blastocystis* might be considered a component of the healthy intestinal microbiota [58,60,71,72,73]. Tito et al. recently surveyed *Blastocystis* subtype prevalence and relative abundance in a Western population cohort. They found that 30% of the healthy population was *Blastocystis* carriers compared to only 9% in an IBD patient cohort, with a significant association between *Blastocystis* subtype and microbiota composition [74]. Moreover, animal studies demonstrate that colonization with *Blastocystis* alters the gut microenvironment in a protective manner, which may, in turn, promote faster recovery from intestinal inflammation [75,76]. The highly contradicting reports between studies describing the impact of *Blastocystis* colonization can be partially attributed to some unexplored variables such as the diversity of intra-subtypes within a subtype, background microbiota composition, host genetics, and diet [77].

Overall, it is becoming evident that the impact of *Blastocystis* colonization on human gastrointestinal health is much more complex than originally envisioned, with the outcome depending on the context and the specific subtype of the protozoa. Thus, it is essential for future studies to elucidate and characterize the contrasting consequences of colonization of different *Blastocystis* subtypes.

### 2.2. Dientamoeba fragilis

Like *Blastocystis* spp., colonization with *Dientamoeba fragilis* has been reported to exert conflicting roles in gut homeostasis. In contrast to other intestinal protozoa whose colonization prevalence is generally considered higher in the emerging nations, *D. fragilis* has been identified more frequently in the developed world [78,79]. However, due to differences in surveillance systems and diagnostic procedures, its prevalence might be underestimated in some regions [78]. The presence of *D. fragilis* has frequently been associated with disease [45], just as it has been commonly found in asymptomatic carriers [72,73,74]. Population-based case-control studies in European countries, including the Netherlands [80], Denmark [60], and Belgium [81], have uniformly reported higher *D. fragilis* colonization in healthy subjects compared to patients with digestive symptoms. Similar to the reported effects of *Blastocystis* colonization, a recent study by Rasmussen et al. showed that colonization with *D. fragilis* is associated with greater gut microbiota diversity compared to non-carriers [82]. Thus far, two subtypes of *D. fragilis* have been identified, with type 1 accounting for the majority of confirmed cases. There are minor phenotypic differences between the two subtypes and their distinction is solely based on genetic diversity. It should be noted that these subtypes have only recently been characterized and many studies have not differentiated between them. It remains unknown whether there is a difference in virulence and pathogenic potential among the two subtypes, which might explain the controversial reports regarding the consequences of *D. fragilis* colonization [45].

### 2.3. Entamoeba spp.

Other common intestinal inhabitants with a worldwide distribution are *Entamoeba* species, the majority of which are generally accepted as commensal organisms [83]. Currently, eight species have been identified that are able to infect humans: *E. histolytica*, *E. bangladeshi*, *E. dispar*, *E. hartmanni*, *E. moshkovskii*, *E. coli*, and *E. polecki*, with *E. histolytica* as the only one with well-established pathogenicity [84]. The worldwide frequency of *Entamoeba* occurrence in humans is estimated at 3.5%. However, the prevalence of commensal *Entamoeba* spp. has largely been underestimated due to high morphological and genetic similarity with the invasive *E. histolytica* [83,84]. Microscopy, the most widely used method for the detection of *Entamoeba* organisms, is not always sufficient for differentiating between the invasive *E. histolytica* and non-pathogenic strains of *Entamoeba* [85]. Increased use of molecular diagnostic methods has recently revealed that colonization with commensal *Entamoeba* spp. is overall more common than infections with *E. histolytica* [85]. A recent cross-sectional study from North India found a higher prevalence of *Entamoeba* spp. in asymptomatic subjects compared to cases with intestinal symptoms, with *E*. *dispar* identified as the predominant species [51]. In agreement with observations made for other commensal protozoa species, individuals colonized with non-pathogenic *Entamoeba* spp. display higher gut microbiota diversity [86]. Additionally, *Entamoeba*-colonized individuals show a shift in the microbiota composition towards a more eubiotic composition, characterized by a decrease of genera associated with autoimmune and inflammatory disease and a corresponding increase of species linked to beneficial effects on intestinal health [87].

Overall, colonization with the above-mentioned protozoa species seems to be of commensal or beneficial nature, especially considering that their colonization is linked to an increase of microbiota diversity, as opposed to infections with pathogenic protozoa species, which often lead to decreased microbiota diversity [88]. However, the contradictory findings between studies support that intestinal protozoa, similar to intestinal bacteria, are a heterogeneous group of organisms that comprises both commensal as well as pathogenic or opportunistic species, with significant variability existing within one species as well, as is the case for *Blastocystis*. Therefore, their importance in human health and disease appears to be largely dependent on the characteristics of a particular species. Furthermore, assessment of the clinical significance of commensal protozoa species in humans relies on appropriate detection methods and accurate species differentiation [37]. DNA-based methods have generally been considered more sensitive for protozoa detection than microscopy, which, until recently, has been the gold standard [89]. On the other hand, DNA-based methods, such as quantitative PCR, can accurately identify the presence of different protozoa species, but fall short in investigating inter-taxa relationships between different microbiota members as well as in determining the relative abundance of a particular species [90]. With recent advances in next generation sequencing strategies, such as e.g., metagenomic approaches, there is an opportunity for greater detection accuracy, detailed taxonomical information, as well as characterization of complex interactions between various microbial populations [37].

## 3. Protozoa–Microbiota Interactions

The various communities of intestinal bacteria play a fundamental role in determining human health. It is generally suggested that high intestinal microbial diversity is a hallmark of a healthy and resilient gut microbiota [91]. Emerging studies consistently report increased bacterial diversity as well as community compositional changes evident in protozoa-colonized individuals (Table 2) [21,22,70,87]. Among the characteristic features of *Blastocystis* colonization is a higher abundance of specific taxa within *Firmicutes*, especially those from the *Clostridia* class, such as *Ruminococcaceae* and *Prevotellaceae* families, and a general decrease of *Bacteroides* abundance [71,74,92]. Furthermore, *Blastocystis* carriers show a significant decrease of *Enterobacteriaceae* and *Proteobacteria* when compared to *Blastocystis*-free subjects [70,71]. Interestingly, *Proteobacteria* and several species within the *Enterobacteriaceae* family can be considered “pathogenic” and linked to microbial dysbiosis associated with the development and pathogenesis of IBD [93,94,95]. Moreover, the presence of *Blastocystis* is strongly associated with the abundance of archaeal organisms, primarily *Methanobrevibacter smithii* [70,71,74]. *M. smithii* has been shown to play an important role in human health, supporting the digestion of glycans through the removal of bacterial fermentation end products [96]. *M. smithii*, together with members of the *Faecalibacterium* and *Roseburia* genera that are also enriched in *Blastocystis*-colonized individuals [22,70], increase the production of the short-chain fatty acid butyrate [97]. Butyrate has well-established beneficial effects on gut health, serving as an important energy source for colonic epithelial cells and acting as an inhibitor of gut inflammation [97,98]. Butyrate-producing bacteria, specifically Faecalibacterium prausnitzii and Roseburia spp., appear to be significantly reduced in patients with Crohn’s disease and have emerged as potential therapeutics for IBD [97,99,100]. Furthermore, a higher ratio of Faecalibacterium prausnitzii to Escherichia coli, as observed in *Blastocystis*- and *Entamoeba*- colonized individuals, has been associated with a healthy and balanced microbial ecosystem [87]. This strongly suggests that colonization with *Blastocystis* selectively reduces the abundance of pathogenic bacteria and induces compositional changes to the microbiota that might be beneficial for the maintenance of intestinal homeostasis.

On the other hand, adverse associations between *Blastocystis* colonization and eubiotic microbial profile have also been described. Several studies have reported a decrease of *Bifidobacterium* in individuals colonized with *Blastocystis* [70,101]. *Bifidobacterium* spp. have been associated with homeostatic functions within the gut, including protection of the epithelial barrier and regulation of inflammation [102]. Accordingly, a study by Alzate et al. showed that children colonized with *Blastocystis* exhibited markedly reduced abundance of the highly beneficial *Akkermansia* spp. compared to children that were *Blastocystis*-free [48]. These results are in line with those recently reported by Caudet et al., who observed decreased abundance of *Akkermansia* spp. and *Bifidobacterium* spp. in obese *Blastocystis*-colonized adults [101]. *Akkermansia* spp. has emerged as one of the important health-promoting microbes, with functions spanning from maintenance of the gut lining to protection against inflammation [103,104]. In contrast to earlier findings, where *Blastocystis* colonization has been associated with a decline of *Lactobacillus* spp. [22], Caudet et al. reported a marked increase of *Lactobacillus* spp. in *Blastocystis*-colonized individuals [101]. A possible explanation for these contradictory findings might be differences in the *Blastocystis* subtype. Of note, the importance of *Blastocystis* subtype characterization when investigating the relationship between *Blastocystis* colonization and gut microbiota composition has been recently emphasized in a study demonstrating an inverse association between *Blastocystis* subtypes and *Akkermansia* spp. relative abundance [74]. *Blastocystis* ST3 showed a negative correlation, whereas *Blastocystis* ST4 was strongly positively correlated to *Akkermansia* and *M. smithii* abundance [74]. Accordingly, *Blastocystis* ST4 was shown in a separate study to ameliorate colonic inflammation in murine colonization models via compositional changes to the gut microbiota that included expansion of *Akkermansia*, as well as modulation of host-immune responses [76]. Furthermore, the presence of *Blastocystis* ST7 in diarrhea patients has been recently shown to be associated with decreased microbiota diversity and bacterial composition changes, characterized by significant enrichment of bacteria belonging to the *Enterobacteriaceae* family as well as *Escherichia coli,* compared to non-infected patients [105]. These results indicate that the impact of *Blastocystis* colonization on gut microbiota composition and structure is subtype-specific, highlighting the importance of *Blastocystis* subtype characterization in future studies.

Research on microbiota composition associated with *D. fragilis* colonization is limited. However, a study conducted in Denmark investigating the microbial profile in *D. fragilis*-positive children revealed 16 bacterial genera that were significantly more abundant in colonized children [82]. Some of the most enriched bacterial genera in *D. fragilis* carriers were *Victivallis*, *Oscillibacter* and *Coprococcus*, whereas *Flavonifractor* was enriched in non-colonized children. After the removal of *D. fragilis* by metronidazole treatment, the abundance of *Flavonifractor* increased while other bacteria, such as *Coproccocus,* were reduced in previously colonized children that were cleared of *D. fragilis*. Evaluating microbiota composition after metronidazole treatment should be done cautiously since this drug is effective against most anaerobic bacteria [106]. However, the microbial heterogeneity caused by metronidazole treatment was transient, with relative abundance reverting to pre-treatment baseline levels within 8 weeks for all genera except for *Flavonifractor*, which kept increasing in children that lost *D. fragilis* [82]. *Flavonifractor* has frequently been associated with disease due to its ability to degrade beneficial anti-carcinogenic flavonoids [107]. Recently, *Flavonifractor plautii* in the intestinal microbiota has been identified as the key bacterium associated with sporadic young-onset CRC [107,108]. Therefore, colonization with *D. fragilis* appears to be linked to beneficial changes in the composition of gut bacteria, thus potentially exerting protective effects against dysbiosis-related diseases.

Colonization with *Entamoeba* spp. results in increased microbiota diversity and compositional changes, characterized by an increase of *Firmicutes* taxa, such as *Ruminococcaceae,* coupled with a significant decrease of *Bacteroides* [21,86]. Interestingly, a reduced ratio of *Firmicutes* to *Bacteroides* causes loss of microbial diversity as well as dysbiosis linked to the progression of IBD, CRC, and type 2 diabetes [109,110]. Recently, it was shown that healthy Colombian children colonized with *Entamoeba coli* exhibit a significant enrichment of *Akkermansia* in their gut microbiota compared to non-colonized children [48]. Moreover, besides a general increase of bacterial richness, a significant enhancement of *Coprococcus* and *Alistipes* was observed in colonized children [48]. *Coprococcus* is an important anaerobic and butyrate-producing bacterium [111], and decreased abundance of *Coprococcus* has been implicated in the pathogenesis of CRC [112]. Furthermore, it was recently demonstrated that *Coproccocus*, as well as other butyrate-producing bacteria, play an important role in language development and cognitive functions in preadolescent children [111].

Together, these studies indicate that commensal gut protozoa significantly remodel the intestinal bacterial niche, potentially creating a favorable microenvironment beneficial for the host. A common observation across the different protozoa species seems to be an enrichment of SCFA-producing bacteria. Importantly, this is in contrast to what has been demonstrated for pathogenic protozoa, e.g., *Cryptosporidium*, where increased infection severity corresponded with a decreased level of fecal SCFA content [113].

However, it is mechanistically unclear how protozoa influence bacterial composition in the gut microbiota. The possibilities include direct modulation e.g., via preferential feeding on specific bacterial species or secretion of metabolites that modulate the fitness of specific bacteria [114,115]. On the other hand, it has been demonstrated that microbiota composition can influence the severity of protozoa infection, as was recently shown for *Cryptosporidium* [113]. This suggests a bidirectional interkingdom crosstalk between different microbiota constituents and supports that background state and composition of intestinal microbiota might play a role in determining the outcomes of commensal protozoa colonization. Additionally, as discussed below in Chapter 4, it is also plausible that changes in bacterial composition are a consequence of direct interactions between the protozoa and the host. Future studies will surely shed more light on the precise mechanisms by which protozoa remodel and are, themselves, influenced by the intestinal bacterial microbiome.

**Table 2 biology-11-01742-t002:** Colonization with different protozoa species and the associated bacterial changes in healthy individuals.

*Protozoa* spp.	Detection Method	Alterations to the Gut Bacterial Microbiota in Colonized Individuals	Ref.	Characteristics of the Enriched Bacterial Species
*Blastocystis*	Real-time PCR	Increase of bacterial genera: *Acetanaerobacterium*, *Acetivibrio*, *Coprococcus*, *Hespellia, Oscillibacter*, *Papillibacter, Sporobacter*, *Ruminococcus*, *Prevotella*, *Roseburia*, *Faecalibacterium* Decrease of bacterial families: *Enterococcaceae, Streptococcaceae*, *Lactobacillaceae* *Enterobacteriaceae*	[22]	*Acetanaerobacterium*: anaerobic, fermentation of acetate ethanol [116] *Coprococcus*: anaerobic, vitamin B, butyrate- and acetate production [111,117] *Hespellia*: anaerobic, butyrate production [118] *Oscillibacter*: anaerobic, glucose oxidation [119] *Papillibacter*: anaerobic, butyrate production [120] *Ruminococcus*: anaerobic, metabolism of complex polysaccharides [121] *Prevotella*: anaerobic, metabolism of polysaccharides [122] *Roseburia*: anaerobic, butyrate-production [120] *Feacalibacterium*: anaerobic, butyrate and other SCFA production [120]
*Blastocystis* *(ST1-6)*	Metagenomics	General increase of *Firmicutes* phyla and *Clostridia* order. Decrease of *Bacteroides* genus Increase of bacterial species: *Methanobrevibacter smithii, Akkermansia muciniphila, Butyrivibrio crossotus, Eubacterium siraeum, Coprococcus catus, Prevotella copri, Eubacterium rectale, Bifidobacterium adolescentis, Faecalibacterium prausnitzii, Treponema succinifaciens* Decrease of bacterial species: *Ruminococcus gnavus, Dialister invisus, Escherichia coli, Bacteroides thetaiotamicron, Bacteroides fragilis, Bacteroides vulgatus, Bacteroides uniformis*, *Bacteroides**ovatus*, *Bacteroides stercoris*	[71]	*Methanobrevibacter*: methanogen, anaerobic, SCFA-production [96] *Akkermansia municiphila*: anaerobic, mucin degrading, SCFA-production [104] *Butyrivibrio crossotus*: anaerobic, butyrate-production [120] *Eubacterium siraeum*: anaerobic, degradation of xylans and ferulic acid production [123] *Coprococcus catus*: anaerobic, SCFA-production [117] *Eubacterium rectale*: anaerobic, SCFA-production [120] *Bifidobacterium adolescentis*: anaerobic, SCFA- and folate production [124] *Treponema succinifaciens*: anaerobic, SCFA-production [125]
*Blastocystis (ST3)*	Microscopic evaluation and real-time PCR	General increase of *Prevotellaceae*, *Methanobacteriaceae, Clostridiaceae Lachnospiraceae, Erysipelotrichaceae and Pasteurellaceae* family. Decrease of *Bacteroidaceae and Veillonecellaceae* family. Increase of bacterial genera: *Prevotella, Methanobrevibacter*, *Ruminococcus* Decrease of bacterial genera: *Bacteroides*	[126]	*Prevotella*: anaerobic, metabolism of polysaccharides [122] *Methanobrevibacter*: methanogen, anaerobic, SCFA-production [96] *Ruminococcus*: anaerobic, metabolism of complex polysaccharides [121]
*D. fragilis*	Microscopic evaluation, multiplex qPCR and real-time PCR	General Decrease of *Bacteroides*. Increase of bacterial genera: *Akkermansia muciniphila, Methanobrevibacter smithii, Butyrivibrio crossotus, Alistipes, Victivallis, Oscillibacter, Eubacterium, Coproccus, Bifidobacterium adolescentis, Bifidobacterium longum, Ruminococcus bromii, Prevotella copri*, Decrease of bacterial genera: *Flavonifractor, Parabacteroides distasonis*, *Bacteroides fragilis,**Clostridium leptum*,	[73,82]	*Methanobrevibacter*: methanogen, anaerobic, SCFA-production [96] *Akkermansia municiphila*: anaerobic, mucin degrading, SCFA-production [104] *Victivallis*: anaerobic and sugar fermenting [127] *Oscillibacter*: anaerobic, glucose oxidation [119] *Coprococcus*: anaerobic, vitamin B, butyrate- and acetate production [111,117] *Bifidobacterium adolescentis*: anaerobic, SCFA- and folate production [124] *Eubacterium siraeum*: anaerobic, degradation of xylans and ferulic acid production [123]
*Entamoeba* spp.	Microscopic evaluation and metagenomics	General increase of taxa *Clostridiales*, *Ruminococcaceae*. Decrease of *Bacteroides*, *Prevotella* and *Fusobacteria* Increase of bacterial genera *Akkarmensia municiphila*, *Coprococcus*, *Alistepes* Decrease of bacterial genera: *Blautia*, *Streptococcus*	[21,48,86]	*Alistepes*: anaerobic, hydrolysis of tryptophan to indole [128] *Coprococcus*: anaerobic, vitamin B, butyrate- and acetate production [111,117] *Akkermansia municiphila*: anaerobic, mucin degrading, SCFA-production [104]
*Entamoeba* and *Blastocystis*	Nested-PCR	Increase of *Faecalibacterium prausnitziim*. Decrease of *Escherichia coli*	[87]	*Feacalibacterium prausnitziim*: anaerobic, butyrate and other SCFA production [120]

## 4. The Impact of Commensal Gut Protozoa on the Host Immune System

Research studies describing the interaction between commensal protozoa species and the mammalian immune system are scarce. Most of the available reports are based on in vitro studies as well as animal models. Although these likely do not fully reflect human (patho)physiology, they still provide valuable insights into potential immunological consequences of intestinal protozoa colonization.

Recently, it was shown that colonization with *Blastocystis* ST4 attenuates colonic inflammation in a dextran sulfate sodium (DSS)-induced colitis mouse model via induction of T helper (Th) 2 cells and T regulatory (Treg) cells [76]. Mice colonized with *Blastocystis* ST4 showed a decrease of tumor necrosis factor-α expressing (TNF) CD4+ T-cells and an upregulation of signature Th2 cytokines interleukin (IL)-4, IL-5, and IL-13, as well as the anti-inflammatory cytokine IL-10 [76]. Additionally, a marked increase of abundance of SCFA-producing bacteria, such as *Ruminococcaceae* and *Roseburia*, was observed following *Blastocystis* ST4 colonization. Analysis of the SCFA content in feces from colitic mice that had received fecal matter transplant from *Blastocystis* ST4-colonized mice revealed enrichment of 6 SCFAs (butyric, isobutyric, valeric, isovaleric, 2-methylbutyric, and caproic acid) compared to mice that received fecal matter transplant from *Blastocystis*-free mice [76]. Importantly, recent reports have repeatedly suggested a highly beneficial role of SCFAs on gut homeostasis and immune modulation [129]. SCFAs in the intestinal lumen are absorbed by colonocytes where they enter the citric acid cycle and are used for energy production. Unmetabolized SCFAs enter the systemic circulation and travel to different organs, serving as substrates or signaling molecules for various cellular processes such as chemotaxis, proliferation, and differentiation [130,131]. SCFAs achieve this by acting as histone deacetylase (HDAC) inhibitors as well as activators of cell surface receptors [130]. It has been demonstrated that butyrate, created by SCFA-producing microorganisms in the intestines, can facilitate generation of extrathymic Tregs via enhanced acetylation of *Foxp3* locus in CD4^+^ T cells. Moreover, butyrate was shown to induce gene expression changes in dendritic cells, characterized by decreased expression of pro-inflammatory cytokines and, consequently, prompting Treg differentiation [132]. Thus, one potential beneficial mechanism of protozoa colonization might be by indirect modulation of the immune system and its skewing towards a Th2/Treg-dominating response via enrichment of SCFA-producing bacteria in the gut microbiota.

A recent in vitro study showed that *Blastocystis* ST4 could decrease the growth of the common pathogen *Bacteroides vulgatus* by inducing reactive oxygen species and the expression of genes associated with oxidative stress. Moreover, a significant reduction in intestinal epithelial permeability was observed in co-cultures with *Blastocystis* ST4, suggesting that this *Blastocystis* subtype protects the intestinal barrier from opportunistic bacterial species, at least in vitro [133]. In agreement with that, it was previously demonstrated that *Tritrichomonas musculis (T. musculis),* a commensal of the rodent microbiota and the closest ortholog to *D. fragilis*, can protect against the mucosal bacterial infection with *Salmonella typhimurium* via induction of intestinal epithelium-derived IL-18 and inflammasome activation. However, due to sustained inflammation characterized by increased numbers of IFN-γ-producing CD4^+^ Th1 and IL-17-producing Th17 cells within the colonic tissue, mice colonized with *T. musculis* were more susceptible to experimental gut inflammation and cancer [16].

On the other hand, *Blastocystis* ST7 has been suggested to have immunocompromising functions, and several potential virulence factors have been identified that could support the notion of pathogenicity. Antigens from *Blastocystis* ST7 have been reported to induce the mitogen-activated protein kinase-dependent expression of pro-inflammatory cytokines such as IL-1β, IL-6, and tumor necrosis factor, in macrophages, mouse intestinal explants, and colonic tissue [134]. Furthermore, *Blastocystis* ST7 has a significantly higher activity of cysteine proteases compared to other *Blastocystis* subtypes. Cysteine proteases are a characteristic feature of parasitic protozoa (e.g., *Entamoeba histolytica* and *Cryptosporidium* spp.) that have been shown to facilitate invasion of host tissue, as well as immune evasion [135]. For instance, *Entamoeba histolytica* utilizes cysteine proteases for degradation of colonic mucins and extracellular matrix components that lead to separation of epithelial cells, consequently breaching the epithelial barrier and enabling the protozoa to invade the host tissue [136,137]. Additionally, cysteine proteases are potent modulators of the host immune defense via direct degradation of immunoglobulin A (IgA), IgG, and IL-18, as well as attenuation of protective Th1-type responses (Figure 1) [135]. Besides *Blastocystis* ST7, no evidence of a similar mechanism has been found in other protozoa species thus far. While *Blastocystis* ST4 did not change the epithelial permeability, *Blastocystis* ST7 induced significant epithelial barrier disruption in epithelial cell lines as well as degradation of IgA [138,139]. Taken together, these data again point toward subtype-specific effects of *Blastocystis* on immune modulation, with ST7 emerging as a major immune-compromising subspecies.

Another important factor that has been often overlooked in experimental studies of protozoa colonization models is the dynamic temporal change that occurs to the microbiota and gut immunity. For example, it has been recently shown that *Blastocystis* ST3 can protect from intestinal inflammation, but only after prolonged colonization time. Short-term exposure experiments, where rat colitis was induced 3 weeks post-colonization with *Blastocystis* ST3, revealed no difference on disease activity; in contrast, long-term exposure (13 weeks post-colonization) resulted in faster recovery and protection from colitis [75]. This might suggest that despite being initially deleterious or neutral, over time, *Blastocystis* ST3 supports a more balanced intestinal microbial ecosystem that is more suitable to control disturbances.

## 5. Conclusions and Future Perspectives

Intestinal protozoa have co-evolved with humans, and their interactions with the human host seem to be highly dynamic and variable, with some species and subtypes exhibiting beneficial properties, while others manifest adverse immunomodulatory effects. The fact that many of these protozoa species cause dormant persistent colonization that often leads to life-long affiliation with their host, points towards commensalism or even symbiosis rather than parasitism. In line with that, colonization with intestinal protozoa appears to significantly increase the diversity of the gut microbiota and selectively modulate the composition of different bacterial communities. Some outstanding questions remain as to whether a therapeutic impact might be achieved by diversification of the human gut via controlled colonization with commensal protozoa strains or by FMT from protozoa-colonized healthy donors to patients with IBD or other gastrointestinal diseases. FMT is an emerging therapy with a successful track record against severe intestinal bacterial infections and a potential therapeutic candidate against diseases associated with microbial dysbiosis [140]. Currently, the presence of protozoa species in human fecal matter donors is an exclusion criterion due to the ongoing debate regarding their pathogenicity. Some preliminary studies in patients subjected to *Blastocystis* transmission have shown no adverse effects in recipients after colonization [140,141]. Investigating whether such modulation of the gut microbiota would be beneficial may be of utmost importance for the development of novel treatment strategies. On the same note, modulation of the host immunity towards a dominant anti-inflammatory response (directly or indirectly through alterations of the bacterial compartment) might constitute an attractive target, especially in relation to autoimmunity. However, more mechanistic studies are needed to decipher the highly complex relationship between commensal protozoa, microbiota, and the host immune responses. While the consensus concerning the role of intestinal protozoa species in health and disease has not been reached, it is becoming increasingly clear that their presence within the gut is not inconsequential to the human host. In-depth characterization of the significance of commensal protozoa colonization might potentially unravel important changes in immunological mechanisms and microbiota dynamics that can greatly benefit our understanding of human intestinal homeostasis.

## Figures and Tables

**Figure 1 biology-11-01742-f001:**
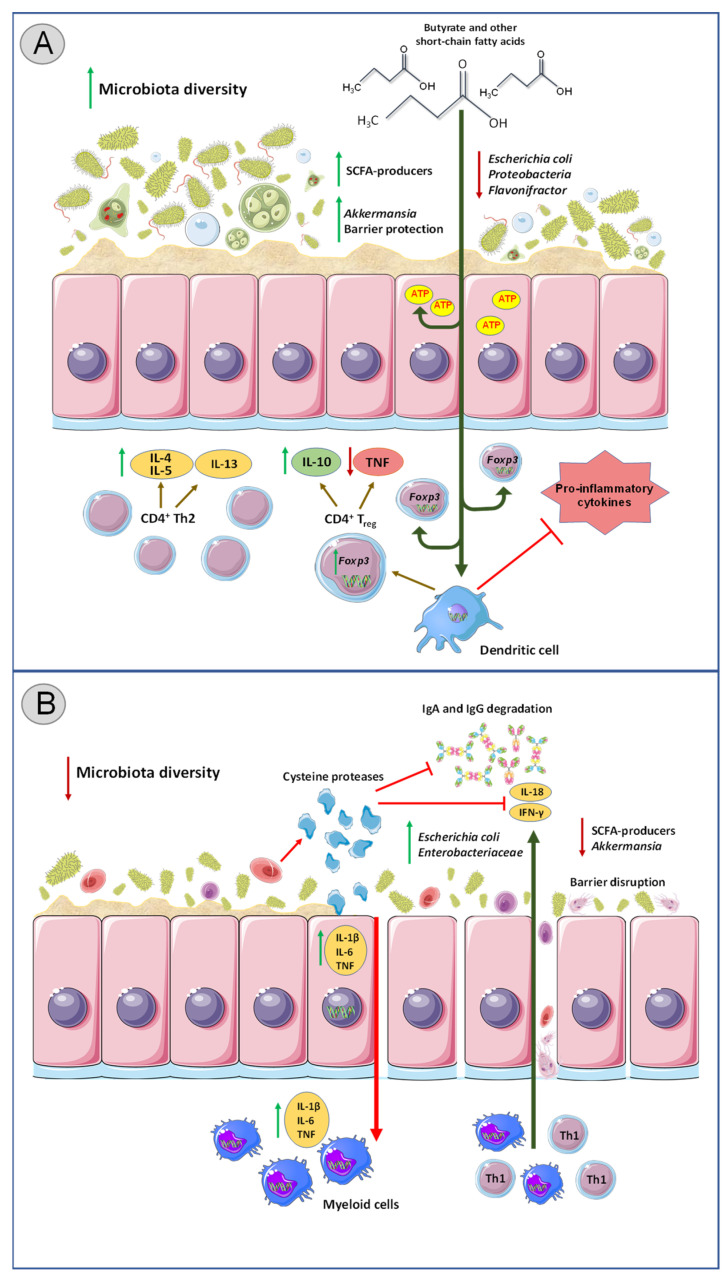
The impact of intestinal protozoa colonization on gut immunity and bacterial composition. (**A**) Colonization with commensal protozoa species (e.g., *Entamoeba* spp. besides *Entamoeba histolytica*) leads to increased microbial diversity and remodeling of bacterial communities in the gut. SCFA (short-chain fatty acids) producers are enriched, and pathogenic bacterial species such as *Escherichia coli* and *Proteobacteria* are reduced. SCFAs in the intestinal lumen are absorbed by epithelial cells for energy production, while unmetabolized SCFAs enter the systemic circulation. Outside the gut, SCFAs facilitate generation of extrathymic Tregs via enhanced acetylation of *Foxp3* locus in CD4^+^ T cells as well as directly affect gene expression in dendritic cells, thus prompting Treg differentiation and consequently IL-10 production. Colonization with commensal protozoa strains leads to polarization of T cell responses towards Th2-dominated profile characterized by IL-4, IL-5, and IL-13 secretion as well as downregulation of TNF. Overall, this results in decreased pro-inflammatory response within the intestines. (**B**) Colonization with pathogenic protozoa species (such as *Cryptosporidium* and *Entamoeba histolytica*) can lead to decreased microbiota diversity, enrichment of pathogenic bacterial species, and a decline in abundance of beneficial bacteria. Cysteine proteases produced by pathogenic protozoa compromise the intestinal epithelial barrier by depleting colonic mucin and create gaps between colonocytes, enabling the parasites to breach the epithelial barrier and invade the tissue. Cysteine proteases also degrade IgA, IgG and IL-18, as well as inhibit Th1-type responses. Pathogenic protozoa also upregulate IL-1B, IL-6, and TNF in the intestines. Overall, these triggers aggravated inflammatory response within the gut. SCFA: Short chain fatty acid, IL: Interleukin, Tregs: T regulatory Th: T helper, TNF: tumor necrosis factor.

**Table 1 biology-11-01742-t001:** Human intestinal protozoa species and their characteristics.

Intestinal Protozoa	Characteristics	Ref.
*Balantidium coli*	Pathogenic ciliates of the human cecum and colon. Causes balantidiosis in humans.	[39]
*Blastocystis hominis*	Questionable pathogenicity. Conflicting effects reported in humans.	[40]
*Chilomastix mesnilii*	Non-pathogenic flagellates of the human large and small intestine. Unknown role and impact on human health.	[41]
*Cystoisospora belli*	Pathogenic coccidians that infects epithelial cells of the intestine. Causes cystoisosporiasis in humans.	[42]
*Cryptosporidium* spp.	Pathogenic coccidians of the small intestine comprising 20 different species identified in humans. Cause cryptosporidiosis in humans.	[43]
*Cyclospora cayetanensis*	Pathogenic coccidians of the human small intestine. Causes cyclosporiasis in humans.	[44]
*Dientamoeba fragilis*	Questionable pathogenicity. Flagellates of the human large intestine. Conflicting effects reported in humans.	[45]
*Endolimax nana*	Non-pathogenic amoebas of the human large intestine. Unknown role and impact on human health.	[46]
*Entamoeba bangladeshi*	Questionable pathogenicity. Amoebas of the human large intestine. Conflicting effects reported in humans.	[47]
*Entamoeba coli*	Non-pathogenic amoebas of the human large intestine. Unknown role and impact on human health.	[48]
*Entamoeba dispar*	Non-pathogenic amoebas of the human large intestine. Unknown role and impact on human health.	[49]
*Entamoeba hartmanni*	Non-pathogenic amoebas of the human large intestine. Unknown role and impact on human health.	[18]
*Entamoeba histolytica*	Pathogenic amoebas of the human intestine. Causes amebiasis in humans.	[50]
*Entamoeba moshkovskii*	Questionable pathogenicity. Amoebas of the human large intestine. Conflicting effects reported in humans.	[51]
*Entamoeba polecki*	Non-pathogenic amoebas of the human large intestine. Unknown role and impact on human health.	[50]
*Giardia intestinalis*	Pathogenic flagellates of the human small intestine. Causes giardiasis in humans.	[35]
*Iodamoeba butschlii*	Non-pathogenic amoebas of the human large intestine. Unknown role and impact on human health	[41]
*Retortamonas intestinalis*	Non-pathogenic flagellates of the human large intestine. Unknown role and impact on human health.	[41]
*Pentatrichomonas hominis*	Questionable pathogenicity. Flagellates of the human large intestine. Unknown role and impact on human health.	[52]

## Data Availability

Not applicable.

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
