# Peer review of "Commensal Intestinal Protozoa—Underestimated Members of the Gut Microbial Community"

_biology, 2022, doi:10.3390/biology11121742_

Round 1

Reviewer 1 Report

In this manuscript, Dubik et al. reviewed the role of protozoa as part of the intestinal microbiota. As this neglected topic is highly complex, this review of the literature can be of great interest for the scientific community. However, in my opinion, the manuscript should be improved before publication.

Major points:

- I strongly suggest to authors to integrate the "technical" aspect of protozoa detection in the manuscript. Indeed, detection of protozoa relies on various methods depending on the studies, with highly variable sensitivity and specificity performance. Thus, it seems critical to know how were detected the protozoa in each study (Table 2). Of note, there is no mention of Next Generation Sequencing (NGS) for detection of eukaryotes, even though this technique seems central for further research on inter-taxa relationships. Globally, I would recommend to be much more critical about the reviewed manuscripts.

Minor points:

- There are numerous mistakes in formatting the names of taxa: only genera and species should be in italics, Giardia lamblia is an old name (prefer G. intestinalis or G. duodenalis) as for Balantidium, Table 1: Chilosmastix mesniliiIodamoeba butschliiCystoisospora belli, Table 2: D. fragilis, L258 Coprococcus...

- L58: "amoebas, flagellates, coccidians and ciliates" is a morphological classification, please write few words on phylogenetic classification

- Table 1: Why is Entamoeba bangladeshi not listed? As for Cryptosporidium other than C. parvum and C. hominis ?

- L170-172: This is partially incorrect, microscopy can differentiate E. coli from E. histolytica/dispar/moshkovskii

- L257: evaluation of microbiota following eradication of D. fragilis by metronidazole treatment seems highly debatable, as 5-nitroimidazole drugs are active against most of anaerobic bacteria

- L296-371: maybe a schematic drawing could highlight main facts about Blastocystis subtypes?

Reviewer 2 Report

This manuscript provides a well-written review of the literature on gut-associated protozoa and their interactions with the bacterial microbiota of the gastrointestinal tract. Pathogenic, non-pathogenic and potentially beneficial species are discussed. Although observations of changes in composition relative to protozoan colonization cannot be definitively related to causality, these relationships do point the direction towards specific interventions and hypotheses that should be tested. As a scientific community, we certainly do need to take protozoa into consideration in modelling the gut microbiota in vitro, especially the lengthy colonization time that appears to be necessary to observe some effects. The variability between and within species is an important point to make for future consideration as well. The section on specific mechanisms for protozoa remodelling of the bacterial community has no references (lines 291-293), while there must be some studies on interactions to support one or the other of the hypotheses. There is a further section on interactions of commensal protozoa with the host immune system that could be referred to for the second mechanism (direct interaction with host). This would improve consistency among sections. One aspect that deserves a bit of attention is the impact of some members of the bacterial/archaea microbiota composition on protozoan infections. This reference may be useful:

https://doi.org/10.1093/cid/ciab207 (Megasphaera in the Stool Microbiota Is Negatively Associated With Diarrheal Cryptosporidiosis)

Specific comments:

Line 83: Some awkward phrasing: “identified to infect” should be inverted to say “seven species able to infect humans have been identified”, or “seven species have been identified that are able to infect humans”.

Line 203: The classification of Enterobacteriaceae as pathogenic in general should be tempered with opposing views of commensal Enterobacteria. Their impact in an autoimmune disease state does not reflect the overall complex relationship of commensal versus pathogenic variants. This is oversimplified.

Line 176, 216-217: Please correct the inconsistent font size.

Line 257: Some species names are not italicized, while joining words such as “and” should not be italicized (line 256).

Table formatting is not optimized. Table 1 has lines within the table (remove them), and the text is centered instead of aligned to the left, making it hard to read.

Table 1: humans have only one small intestine each, so the word should be singular, not plural. If referring to the separate sections, then specific nomenclature should be used.

Table 2: double-spaced lines are not consistent with the rest of the text. Remove right-left justification on text. It may be better to have three columns, one for the main reference, but dividing up the increased genera and decreased families/species. References in square brackets should be spaced from the text (both left and right). Some species names are written with a capital letter; should be lower case.

Lines 291-293: Are there any references that can be given for preliminary study components on possible mechanisms?

References: the formatting is not consistent (titles and journal names)

Round 2

Reviewer 1 Report

Modifications were done accordingly to my suggestions, or consistently answered.